# Pre-Training Representations of Binary Code Using Contrastive Learning

**Yifan Zhang**                                                    *yifan.zhang.2@vanderbilt.edu*
*Department of Computer Science*
*Vanderbilt University*

**Chen Huang**                                                    *huang__chen@nus.edu.sg*
*Department of Computer Science*
*National University of Singapore*

**Yueke Zhang**                                                   *yueke.zhang@vanderbilt.edu*
*Department of Computer Science*
*Vanderbilt University*

**Huajie Shao**                                                   *hshao@wm.edu*
*Department of Computer Science*
*College of William & Mary*

**Kevin Leach**                                                   *kevin.leach@vanderbilt.edu*
*Department of Computer Science*
*Vanderbilt University*

**Yu Huang**                                                      *yu.huang@vanderbilt.edu*
*Department of Computer Science*
*Vanderbilt University*

**Reviewed on OpenReview:** [https://openreview.net/forum?id=qmfUL6D0iz](https://openreview.net/forum?id=qmfUL6D0iz)

## Abstract

Binary code analysis and comprehension is critical to applications in reverse engineering and computer security tasks where source code is not available. Unfortunately, unlike source code, binary code lacks semantics and is more difficult for human engineers to understand and analyze. In this paper, we present CONTRABIN, a contrastive learning technique that integrates source code and comment information along with binaries to create an embedding capable of aiding binary analysis and comprehension tasks. Specifically, we present three components in CONTRABIN: (1) a primary contrastive learning method for initial pre-training, (2) a simplex interpolation method to integrate source code, comments, and binary code, and (3) an intermediate representation learning algorithm to train a binary code embedding. We further analyze the impact of human-written and synthetic comments on binary code comprehension tasks, revealing a significant performance disparity. While synthetic comments provide substantial benefits, human-written comments are found to introduce noise, even resulting in performance drops compared to using no comments. These findings reshape the narrative around the role of comment types in binary code analysis. We evaluate the effectiveness of CONTRABIN through four indicative downstream tasks related to binary code: algorithmic functionality classification, function name recovery, code summarization, and reverse engineering. The results show that CONTRABIN considerably improves performance on all four tasks, measured by accuracy, mean of average precision, and BLEU scores as appropriate. CONTRABIN is the first language representation model

to incorporate source code, binary code, and comments into contrastive code representation learning and is intended to contribute to the field of binary code analysis. The dataset used in this study is available for further research.

# 1 Introduction

Binary code[1] provides valuable information about a program's content and behavior and is often the only available representation of a program in certain cases, such as legacy systems, proprietary software, and penetration testing (Harris & Miller, 2005). Binary code analysis is critical to many software related tasks, and understanding binary code can serve as a critical source of information about program content and behavior, and thus a foundation of many applications (Wang et al., 2000; Pierce & Mudge, 1994; Cifuentes et al., 1999; Giffin et al., 2004).

However, while understanding and analyzing binary code is essential for various tasks, there exists no practical, generalizable strategy for comprehending binary code due to inherent characteristics: compared to source code and natural languages, binary code has very limited semantics, is substantially more difficult for humans to understand, and is much more difficult to analyze automatically.

Currently, binary code analysis is approached through two methods: (1) traditional static and dynamic analysis and (2) AI-based methods. Traditional methods use manual techniques and specific algorithms (e.g., dataflow analysis), but are limited in their ability to be used across different platforms and applications (Bao et al., 2014; David et al., 2016; Aslanyan et al., 2020). AI-based methods use machine learning to capture both program structure and semantics, and have combined syntax, semantics, and structure to create embeddings for specific tasks (Guo et al., 2020; Lacomis et al., 2019; Chen et al., 2022b). However, few focus on large-scale pre-trained representations for binary code, which is a largely unexplored area (Feng et al., 2020; Guo et al., 2020), instead focusing primarily on source code or related syntactic structures like abstract syntax trees.

Considering the critical challenges needed to assist binary code analysis, in this paper, we present CONTRA-BIN, a novel contrastive learning model that combines simplex interpolation to approximate human gradual learning, and intermediate contrastive learning to produce a high quality embedding for binary code. Large corpora of source code and documentation can be combined with corresponding compiled binary code to serve as a basis for an effective contrasting learning approach.

CONTRABIN consists of three components and an evaluation, illustrated in Figure 2. First, we leverage comments within the source code and the corresponding binary code (which can be obtained from compiling the source code), and randomly choose two of the three representations (i.e., source code, comments, and binary code) to conduct primary contrastive learning. Then, we design linear and non-linear simplex interpolation methods to interpolate two data embeddings and obtain an intermediate representation. Next, we introduce an intermediate contrastive learning approach to incorporate source code and comment information into binary code. Lastly, we evaluate the trained binary code embedding on three downstream tasks for binary code analysis to validate our model. We elaborate upon the design of CONTRABIN in Section 2.

The design of CONTRABIN is based on three key insights. First, we recognize that source code, comments, and their compiled binaries are different but semantically equivalent representations of a program, forming a rich multi-modal view. Second, we observe that the process of a human documenting code and a compiler translating it share properties of gradual learning, as illustrated in Figure 1. We model this with simplex interpolation to create a unified learning process that bridges these different views. Finally, our approach leverages the principle of semantic dissimilarity: while these related artifacts represent the same program, they are inherently distinct from the artifacts of any other program. These insights form the foundation of our multi-modal contrastive learning framework, designed to produce effective embeddings for binary analysis.

---

[1]In this paper, we use *binary code* to refer to both executable binary functions and lifted assembly/intermediate representations because they are trivially interchangeable.

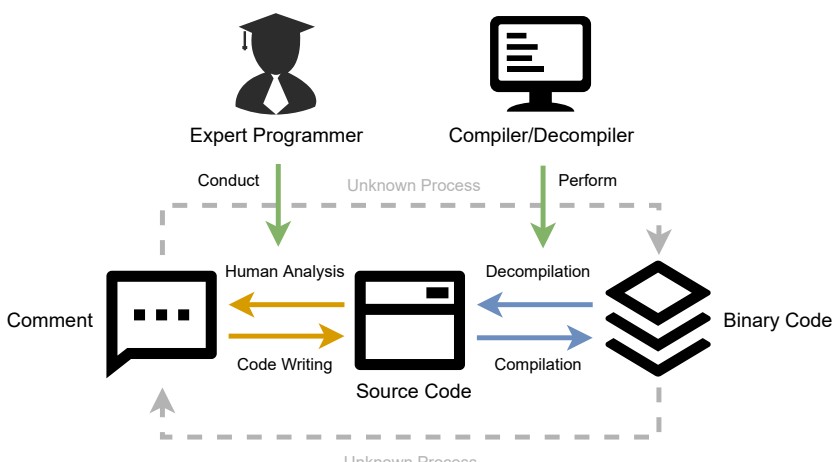

Figure 1: Illustration of the software compilation and human documentation process. Expert programmers can analyze source code to write comments or produce code based on comments. Compilers can compile source code and obtain binary code, whereas decompilers can decompile binary code and get source code.

To validate our approach, we evaluate CONTRABIN's pre-trained embeddings against several state-of-the-art large-scale models (Liu et al., 2019; Feng et al., 2020; Guo et al., 2020; 2022a). While many existing models are designed primarily for source code, we make a best-effort comparison to establish a strong baseline for binary-focused representation learning. Our evaluation spans four indicative downstream tasks: (1) Binary Functional Algorithm Analysis on the POJ-104 dataset (Mou et al., 2016); (2) Binary Function Name Recovery on the DIRE dataset (Lacomis et al., 2019); and (3) Binary Code Summarization and (4) Binary Reverse Engineering, both on subsets of AnghaBench (Da Silva et al., 2021).

These tasks were chosen as they provide a comprehensive test of binary analysis and comprehension capabilities and permit a fair comparison with source-code-aware models. Our results show that CONTRABIN substantially outperforms current code analysis models on the vast majority of task-relevant metrics and achieves comparable performance on others.

In conclusion, we claim the following contributions:

- We present CONTRABIN, the first contrastive learning framework, to our knowledge, for learning representations of LLVM Intermediate Representation (IR). It uniquely aligns three modalities (source code, IR, and comments) using a two-stage process with simplex interpolation to create meaningful intermediate views.
- We design a novel simplex interpolation method, inspired by multi-view and curriculum learning, that models a gradual learning process to better align the high-level semantics of source code and comments with the low-level structure of binary code.
- We provide a key empirical finding on training data quality: concise, LLM-generated comments significantly improve performance, while noisy, human-written comments degrade it, reshaping how natural language annotations should be leveraged in binary analysis.
- Our extensive evaluation on four diverse tasks—functionality classification, name recovery, summarization, and reverse engineering—shows that CONTRABIN consistently and significantly outperforms strong pre-trained baselines, demonstrating the broad applicability of its embeddings.

Collectively, these contributions are embodied in CONTRABIN, a framework designed to significantly advance the state of binary code representation learning. By bridging the semantic gap between source code, natural language, and low-level binaries, our work provides a robust foundation for a wide range of security and reverse engineering applications. To facilitate further research and ensure the reproducibility of our results, we have made our complete implementation, pre-trained models, and all datasets publicly available.[2]

---

[2]https://zenodo.org/records/15219264

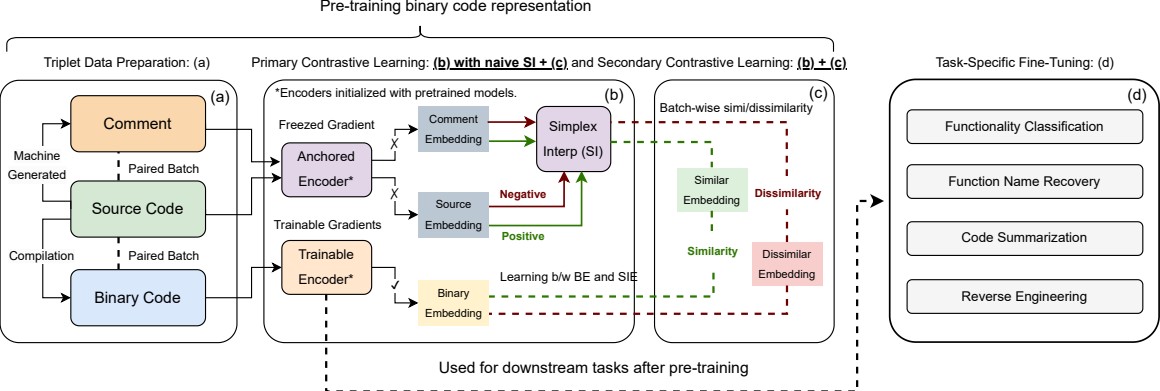

Figure 2: Overview of CONTRABIN's Training Framework. The framework consists of four components: (a) **Triplet Data Preparation**, where triplet data (source code, binary code, and comments) are created through compilation and machine-generated comments; (b) and (c) are collectively structured into two learning stages: (1) **Primary Contrastive Learning**, which selects a single representation (naive SI) and applies batch-wise similarity and dissimilarity; and (2) **Secondary Contrastive Learning**, which generates interpolated representations using simplex interpolation (SI) and further aligns them. Finally, (d) **Task-Specific Fine-Tuning** applies the pre-trained embeddings to downstream tasks.

## 2   Approach

In this section, we describe CONTRABIN, our approach to obtain high quality representation for binary code that can be used for a swath of binary analysis and comprehension tasks. In practice, our goal is to augment stripped, semantics-dearth binaries with rich contextual information provided by comments and source code. At a high level, we use a large-scale pre-training task along with downstream tasks related to binary code. CONTRABIN leverages tuples of (source code, binary code, comments) from many different programs scraped from GitHub, which form the basis of a contrastive learning task. The architecture of CONTRABIN is summarized in Figure 2, which consists of three components. Given source code, we first generate comments using pre-trained models, compile each code snippet to emit binary code, and randomly choose two of the three code representations in each batch of triplets to conduct primary constrastive learning. Then, we derive embeddings for the triplets from our encoder as a simplex projection, and generate an intermediate representation based on random and learnable interpolation. Next, we train our encoder by intermediate contrastive learning using anchored representation and intermediate representation (i.e., one of the three representations is anchored while the other two are not). Finally, we apply the trained binary code embedding to four downstream tasks to evaluate its performance.

**Background** In recent years, pre-trained representations have become a powerful tool in code analysis, offering the ability to leverage vast amounts of existing knowledge to improve the performance and generalization of models. By pre-training on large datasets of source code and natural language comments, these representations capture essential patterns and semantics that are difficult to learn from binary code alone. In our approach, we utilize these pre-trained models to bridge the gap between the rich, high-level information available in source code and comments, and the lower-level, more abstract nature of binary code. This strategy allows our model to develop a deeper understanding of binary code by learning from multiple modalities, ultimately enhancing its ability to perform complex tasks in code analysis.

To achieve this, our approach is structured in several key stages, each building on the previous one to refine and improve the model's understanding of binary code. We begin by aligning the representations of source code, comments, and binary code through a series of pre-training steps designed to capture the nuances of each modality. By progressively refining these embeddings, our model learns to effectively transfer knowledge between different forms of code representation, ensuring that the final binary code embedding is robust and

semantically rich. The following sections detail each of these stages, illustrating how they contribute to the overall effectiveness of our methodology.

## 2.1 Primary Contrastive Learning

This subsection introduces the primary contrastive learning approach for the simplex interpolation. Recall that the conversion process between two representations can involve complex analyses (e.g., compilation) or creative human processes (e.g., code comprehension). At the same time, no generalizable models have been trained on datasets containing binary code. This can lead to the cold start problem in representation learning (Contardo et al., 2014) when a new data representation (e.g., a novel program never seen before) emerges in a system. Therefore, it is challenging to capture their features — the same scenario applies to binary code representation.

> **Comparison** The initial phase of CONTRABIN reflects the early learning stages individuals go through when encountering a new concept, emulating it using straightforward comparisons.

To mitigate this cold start problem, we adopt primary contrastive learning in the first step of model training. Specifically, we use three representations of a program: comments (i.e., summarization generated by code models), source code (i.e., as written by developers), and binary code (i.e., as emitted from a compiler), and randomly choose two of them to conduct contrastive learning, as shown in Figure 2. During this step, the simplex interpolation module is disabled:

this approach is inspired by multi-view learning (Zhao et al., 2017) in computer vision, in which an object can have $k$ views, and all of the views are exactly from the same object. In our model, either source code, binary code, or comment can represent the same program, so we treat the three representations (also denoted as *modalities* or *views*) of a program to enrich the information in binary code

There are two steps in primary contrastive learning: manifold projection and batch-wise similarity comparison.

**Manifold projection** To perform primary contrastive learning, the vector representation for each input string (i.e., of instructions, source code, or comments) is obtained by an encoding function $f_\mathcal{M}$ that projects an instance $x$ into a manifold space $\mathcal{M}$ with dimension $d$ In this paper, $x_s$, $x_b$ and $x_c$ denote input string of source code tokens, binary code (assembly or IR lifted from a binary), and comment of one program, while $h_s, h_b, h_c \in \mathcal{R}^d$ are their corresponding vector representations. The batch-wise manifold projection is

$$
\begin{aligned}
H_s &= f_\mathcal{M}^a(X_s), \quad H_c = f_\mathcal{M}^a(X_c), \\
H_b &= f_\mathcal{M}^t(X_b),
\end{aligned}
\tag{1}
$$

where $X_s$, $X_b$ and $X_c$ are the matrix representations of a batch of programs, and $H_s$, $H_b$ and $H_c \in \mathcal{R}^{n \times d}$ are the matrix representations of a batch of projected embedding. We use a single anchored encoder $f^a$ for encoding source code and comments and a trainable encoder $f^t$ for binary code. In this context, the use of the terms 'anchored' and 'trainable' refers to the fact that the parameters of the encoder used for encoding comments and source code will not be updated by any binary code. Only the binary code encoder is trainable. This design was made based on our early experimental findings, in which we found that using binary code to update comments and source code resulted in an ineffective model due to the considerable differences between binary code and these other forms of representation. To optimize the training process and build upon existing knowledge, we utilized pretrained embeddings, allowing for more efficient adaptation to the task of binary code representation. This strategy reduces the need for training from scratch and enables a more effective fine-tuning process.

**Batch-wise similarity comparison** To align binary code representations with their corresponding source code and comments, we adopt a loss function inspired by CLIP (Radford et al., 2021). This method leverages batch-wise similarity and dissimilarity between different program representations to update the model.

Specifically, we compute the in-batch cross-similarity between the anchored representation (either comment or source code) and the trainable binary code representation as follows:

$$\text{logits} = H_{c/s}H_b^T, \tag{2}$$

where $H_{c/s}$ and $H_b$ represent the embeddings of comments or source code and binary code, respectively, and the notation $H_{c/s}H_b^T$ indicates standard matrix multiplication. The resulting logits quantify the in-batch cross-similarity.

To ensure compatibility with the cross-entropy loss, we convert the logits into a probability distribution using the softmax function:

$$P = \text{softmax}(\text{logits}, \dim = -1). \tag{3}$$

Next, we compute the in-batch similarity of the representations within each modality:

$$\text{sim}_{c/s} = H_{c/s}H_{c/s}^T, \quad \text{sim}_b = H_bH_b^T, \tag{4}$$

where $\text{sim}_{c/s}$ and $\text{sim}_b$ represent the intra-batch similarities of the comment/source code and binary code embeddings, respectively.

To construct the target distribution for the loss computation, we take the average of the intra-batch similarities and apply the softmax function:

$$\text{targets} = \text{softmax}\left(\frac{\text{sim}_{c/s} + \text{sim}_b}{2}, \dim = -1\right). \tag{5}$$

The loss function is then computed using the cross-entropy (CE) between the probability distributions of $P$ and targets:

$$\mathcal{L} = CE(P, \text{targets}). \tag{6}$$

By minimizing this loss, the model learns to align semantically similar binary code, source code, and comment embeddings in a unified embedding space. Minimizing the cross-entropy penalizes differences between $P$ and targets, forcing the two distributions to become more similar. As targets reflects the similarity structure of embeddings, this process aligns $P$ with it, effectively concentrating the distributions (reducing entropy) and pulling semantically related concepts closer in the embedding space.

**Explanation of the loss function** The loss function is designed to address the inherent differences between binary code and higher-level code or comments. By minimizing this loss, the model learns to position related concepts—whether from binary code, source code, or comments—closely together in a unified embedding space. This approach enhances the model's ability to transfer knowledge across these diverse representations, leading to improved performance in downstream tasks. Although inspired by the contrastive loss used in CLIP, this function is tailored to the specific needs of binary code representation, ensuring that similar embeddings are aligned while distinct ones are separated. This loss function can then help to align the binary code embedding closer to the representations of comments and binary code in the manifold $\mathcal{M}$, thereby providing a well-trained binary code embedding as the starting point for subsequent steps. The model will then back propagate the loss and update the parameters.

## 2.2 Secondary Contrastive Learning

To enhance the representation quality of binary code embeddings, secondary contrastive learning builds on primary contrastive learning by integrating simplex interpolation and intermediate contrastive learning. In primary contrastive learning, naive interpolation is used, where a single representation is selected without applying interpolation techniques (as illustrated in Figure 2). Secondary contrastive learning extends this by introducing two advanced simplex interpolation methods: linear interpolation for direct learning and non-linear interpolation for customized learning. These methods generate diverse and expressive views of program representations, enabling the model to better capture semantic relationships among source code, binary code, and comments. Intermediate contrastive learning then operates across all views generated by

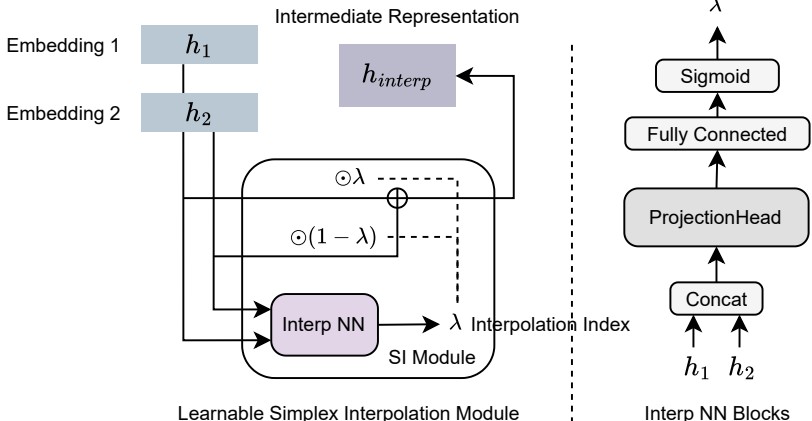

Figure 3: Illustration of the Simplex Interpolation Process. The left side shows our linear interpolation pipeline, learning a $\lambda$ to interpolate $h_1$ and $h_2$, two of the $h_c$, $h_s$ and $h_b$. The right side provides one type of interpolation NN block, consisting of the structure of ProjectionHead, a fully-connected layer, and the final Sigmoid layer.

the simplex interpolation methods, providing a novel mechanism to align semantically related representations while preserving distinctions among unrelated ones. Figure 2 illustrates the integration and interaction of these components within a unified training pipeline.

**Simplex interpolation for gradual learning** After completing primary contrastive learning, we use simplex interpolation for gradual learning among the three representations, as shown in Figure 2 (a). Simplex interpolation is inspired by human translation (e.g., from source code to comment or vice versa) and the code compilation (or decompilation) process, in which conversion from one representation to the other involves intermediate thinking or analysis. While we acknowledge that the manifold assumption is a common tool in understanding deep learning phenomena, its application here serves primarily to align with our color boxes discussed from Section 2.1 to Section 2.4. This assumption provides a simplified framework to conceptualize the interpolation process, aiding in the explanation of our model's behavior in binary code representation learning.

> **Analogy and extrapolation** The subsequent portion suggests that CONTRABIN compares representations of the same program to learn differences and similarities among programs.

We use both linear and non-linear simplex interpolation as part of a gradual learning approach to obtain increasingly expressive embeddings. Based on simplex interpolation theories (Thrampoulidis et al., 2022), we presume the intermediate representations between any of the two in source code, binary code, and comments can be a different view of the same program. The detailed steps are shown as follows:

**Linear interpolation for direct learning** We create two interpolation methods to generate inputs for contrastive learning: linear interpolation and non-linear interpolation, as shown in a unified Figure 3. Both interpolation methods follow the same interpolation function $\Gamma(A, B; \lambda)$ defined as:

$$\Gamma(A, B; \lambda) = \lambda \cdot A + (1 - \lambda) \cdot B, \tag{7}$$

where $A$ and $B$ are batch data representations, $\lambda \in [0, 1]$ denotes the interpolation index, and $\cdot$ indicates element-wise multiplication. In CONTRABIN, we define $H_l = \Gamma(H_1, H_2; \lambda_l)$ and $H_n = \Gamma(H_1, H_2; \lambda_n)$ for linear and non-linear interpolation, respectively. Linear interpolation imitates the learning process between one thing to another, where the model progresses towards all knowledge points for a given task. In practice, we regard the interpolation index as a trainable parameter, learned by a neural network, denoted as:

$$\lambda_l = f_{interp}^l(H_1 + H_2), \tag{8}$$

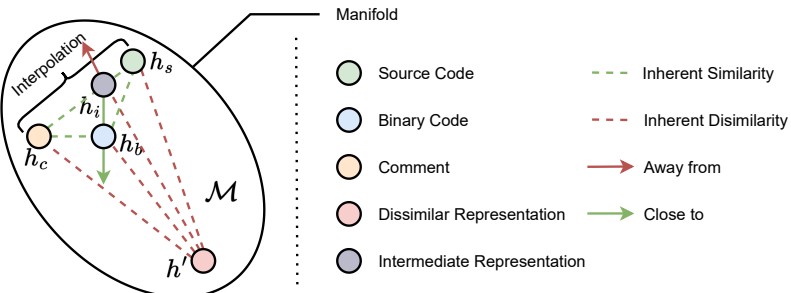

Figure 4: Illustration of the intermediate contrastive learning. The source code $h_s$, binary code $h_b$, comment $h_c$, and their interpolations $h_i$ in manifold $\mathcal{M}$ are inherently similar, whereas they are dissimilar to any other program representations $h'$. Therefore, they will be close to one another but away from others during training.

where $f_{interp}^l$ denotes the interpolation neural network function that learns the optimal interpolation index for $H_1$ and $H_2$, and $\lambda_l$ denotes a linear interpolation index ranging from 0 to 1. Note that the Interp NN blocks in Figure 3 are one of the possible implementations. We apply linear interpolation to the manifold vectors of source code and binary code using:

$$H_l = \lambda_l \cdot H_1 + (1 - \lambda_l) \cdot H_2, \tag{9}$$

where $H_l$ denotes the batch-wise intermediate representation generated by the linear interpolation. The broadcast property is adopted so that the scalar $\lambda_l$ expands to the shape of $H_1$ and $H_2$, enabling element-wise multiplication.

**Non-linear interpolation for customized learning** We further consider non-linear interpolation to capture complex semantics and enhance model performance. Unlike linear interpolation, which uses a scalar $\lambda$, non-linear interpolation treats the interpolation index as a matrix with the same shape as $H_1$ and $H_2$, allowing finer-grained and feature-wise control. Formally, this is defined as:

$$\lambda_n = f_{interp}^n(H_1 + H_2), \tag{10}$$

$$H_n = \lambda_n \cdot H_1 + (1 - \lambda_n) \cdot H_2, \tag{11}$$

where $\lambda_n$ is a matrix representing the non-linear interpolation index and $f_{interp}^n$ is the corresponding neural network function. Both interpolated embeddings $H_l$ and $H_n$ serve as input to the intermediate contrastive learning step.

**Intermediate contrastive learning for representation refinement** To further refine the binary code embeddings, we introduce intermediate contrastive learning, which leverages interpolated representations generated in Subsection 2.2. These interpolations provide an alternative view of the program's semantics, acting as unseen yet reasonable information about the same program. This step builds on the concept of contrastive learning by incorporating these interpolated representations into the training process.

As illustrated in Figure 4, the interpolated representation $h_i$ (a combination of source code projection $h_s$ and comment projection $h_c$) is compared against the binary code projection $h_b$ in the embedding manifold $\mathcal{M}$. Similarly, $h'$ denotes all dissimilar projections, which include other representations in the batch that do not belong to the same program.

To compute the intermediate contrastive loss, we define

$$Z_i = \sum_{h' \in \mathcal{B}'} \exp(\text{sim}(h_i, h')/\tau),$$

---

**Algorithm 1** CONTRABIN pre-training framework

---

**Require:** Source code set $X_s$ with paired binary code set $X_b$ and comment set $X_c$, split into train $X^{train}$ and validation $X^{val}$.

**Ensure:** Minimization of loss on $X^{val}$.

1: **if** Primary contrastive learning **then**
2:     **for** $batch = 1, \ldots, k_{start}$ **do**
3:         Project $X_c$, $X_s$ and $X_b$ to $\mathcal{M}$ with encoders $f_a$ and $f_t$
4:         Compute batch-wise similarity loss between two representations in $\mathcal{M}$ and train the predictors and encoder
5:     **end for**
6: **end if**
7: **if** Contrastive learning by linear **or** non-linear simplex interpolation **then**
8:     **for** $batch = 1, \ldots, k_{interp}$ **do**
9:         Project $X_c$, $X_s$ and $X_b$ to $\mathcal{M}$ with encoder $f_a$ and $f_t$
10:         Choose two of the encoded representations and compute the index $\lambda_l$ or $\lambda_n$ by the interpolation NN and get intermediate representations
11:         Compute batch-wise similarity loss between intermediate and the other representation in $\mathcal{M}$ and train the binary encoder and interp NN
12:     **end for**
13: **end if**

---

where $\mathcal{B}'$ denotes the set of all dissimilar projections in the batch, $\text{sim}(\cdot, \cdot)$ represents the cosine similarity between two embeddings, and $\tau$ is a temperature parameter. The loss is then computed as:

$$\mathcal{L}_{\text{intermediate}} = -\frac{1}{N} \sum_{i=1}^{N} \log \left( \frac{\exp(\text{sim}(h_i, h_b)/\tau)}{\exp(\text{sim}(h_i, h_b)/\tau) + Z_i} \right),$$

where $N$ is the batch size. This formulation encourages the interpolated representation $h_i$ to align closely with the corresponding binary code projection $h_b$, while maintaining separation from unrelated projections $h'$. By introducing $Z_i$, we explicitly represent the normalization factor in the denominator, which simplifies the interpretation of the loss and enhances its clarity.

> **Intermediate Representations in Context** Intermediate contrastive learning refines embeddings by aligning related representations and separating unrelated ones, enabling CONTRABIN to better capture semantic relationships for improved learning.

To stabilize training, we adopt a stop-gradient mechanism for the anchored representation during optimization, detaching the gradient of $h_s$ or $h_c$ when used as the anchor to prevent instability caused by intermediate representations. This mechanism facilitates the integration of intermediate contrastive learning into the overall training process, ensuring that interpolated views contribute effectively to refining binary code embeddings. Building on this foundation, the subsequent gradual learning step further enhances the model's ability to capture nuanced semantic relationships across diverse program representations.

**Gradual learning** Following the gradual learning literature, models learn most effectively when they follow a natural, progressive order as humans do (Gou et al., 2021; Cho & Hariharan, 2019). Specifically, models learn high-level data abstracts initially, then progress to more complicated embeddings. Following this intuition, we train our model via primary contrastive learning during cold start. After several epochs (10 epochs in our implementation), we switch the training pattern to contrastive learning based on linear simplex interpolation and non-linear simplex interpolation to further improve the generalizability of our model and finalize the training process, shown in Algorithm 1. By integrating all three parts, CONTRABIN follows a natural learning curve (i.e., from simple to complex) to better understand the meaning of binary code.

Table 1: Categorization and objectives of downstream tasks.

| Task | Domain | Objective |
|---|---|---|
| Algorithmic functionality classification (1) | Analysis | Categorize binaries by functionality |
| Function name recovery (2) | Analysis | Predict meaningful function names |
| Code summarization (3) | Comprehension | Generate concise summaries for binaries |
| Reverse engineering (4) | Comprehension | Decompile binaries into source code |

### 2.3 Task-Specific Fine-Tuning

To evaluate the effectiveness of our pre-trained binary code embeddings, we apply them to four downstream tasks: (1) **algorithmic functionality classification from binaries**, (2) **binary code function name recovery**, (3) **binary code summarization**, and (4) **binary reverse engineering**. These tasks comprehensively span two domains: binary code analysis (tasks 1 and 2) and binary code comprehension (tasks 3 and 4). By addressing a diverse set of challenges, our fine-tuning approach demonstrates the versatility and robustness of the learned embeddings.

Fine-tuning is performed using task-specific objectives, each tailored to the requirements of the respective downstream task. For classification tasks, including algorithmic functionality classification and function name recovery, we employ a cross-entropy loss function:

$$\mathcal{L}_{\text{classification}} = -\frac{1}{N} \sum_{i=1}^{N} y_i \log \hat{y}_i,$$

where $y_i$ is the true label, $\hat{y}_i$ is the predicted probability, and $N$ is the batch size. This objective encourages the model to predict accurate labels for binary code representations.

For summarization and reverse engineering tasks, we treat the problem as a sequence-to-sequence generation task. Here, the loss function used is a token-level cross-entropy loss:

$$\mathcal{L}_{\text{seq2seq}} = -\frac{1}{N} \sum_{i=1}^{N} \sum_{t=1}^{T} y_{i,t} \log \hat{y}_{i,t},$$

where $T$ represents the sequence length, $y_{i,t}$ is the true token at time step $t$, and $\hat{y}_{i,t}$ is the predicted probability for that token. This formulation ensures the model learns to generate accurate textual summaries and source code representations.

All four tasks are adapted from the CodeXGlue[3] benchmark, with modifications for binary code analysis. For instance, the binary functionality classification and function name recovery tasks are based on their corresponding tasks in CodeXGlue but focus on pre-compiled binary representations rather than source code. Similarly, binary code summarization and reverse engineering extend the summarization and translation tasks in CodeXGlue to the domain of binary-to-natural language and binary-to-source code transformations, respectively. Detailed descriptions for each task are provided below.

**Algorithmic Functionality Classification from Binaries** This task involves classifying binary code according to its functionality, such as distinguishing "quicksort" from "md5 hash." Accurate classification assists developers in isolating specific code blocks, improving readability, saving development time, and aiding in reverse engineering (Haq & Caballero, 2021). For example, given a binary, the goal is to categorize it into classes like sorting, hashing, or searching. As shown in Table 1, this task is categorized under "Analysis" and represents a key aspect of binary code analysis, where our method leverages multiple program representations for robust classification.

**Function Name Recovery** Recovering function names from binary code sequences is crucial for stripped binaries that lack meaningful names and debugging information. This task, categorized as "Analysis" in Table 1, significantly reduces manual effort in reverse engineering and malware analysis scenarios, particularly

---

[3]https://github.com/microsoft/CodeXGLUE

when source code is unavailable. By predicting function names effectively, our embeddings restore clarity to binary code, enhancing the overall analysis workflow.

**Code Summarization** This task translates binary code into concise English summaries, aiding comprehension and facilitating security assessments, reverse engineering, and software maintenance. As outlined in Table 1, code summarization falls under the "Comprehension" category, emphasizing its role in improving understanding of binary functionality and enabling more effective software analysis through succinct summaries.

**Reverse Engineering** Decompiling binary code into source code is essential for tasks like security assessments, defect localization, and software comprehension. While tools like Hex-Rays and Ghidra offer decompilation support, they often generate code that lacks symbolic names and deviates from canonical developer-written formats. Reverse engineering, classified under "Comprehension" in Table 1, benefits from our embeddings by transforming binaries into more interpretable source code, providing valuable insights into the original binaries.

We summarize the categorization of these tasks, their objectives, and their respective focus areas in Table 1, which groups the tasks into "Analysis" and "Comprehension" domains.

### 2.4 Parallelism Between the Presented Steps and the Learning Stages

To ensure a clear understanding of the connections between the training steps and the overall learning pipeline, we explicitly describe the parallelism between the training stages and learning steps in CONTRABIN. Each stage in our framework directly contributes to a specific aspect of the learning process, as illustrated in Figure 2:

- **Primary Contrastive Learning Step**: This step corresponds to the initial stage of representation learning, where naive interpolation (selecting a single representation from source code or comments) is used. It serves as the foundational phase to align binary code with these modalities in the embedding space without introducing intermediate representations.

- **Secondary Contrastive Learning Step**: This step builds upon the foundational alignment by introducing simplex interpolation, where representations of source code and comments are interpolated to generate intermediate views. These interpolated representations provide a nuanced semantic bridge, which, when combined with intermediate contrastive learning, allows the model to refine its embedding space. By focusing on relationships between source code and comments, this step ensures that binary code embeddings are enriched through contextually meaningful relationships.

- **Task-Specific Fine-Tuning Step**: In this final stage, the pre-trained embeddings are applied to downstream tasks. This step translates the learned embeddings into practical utility, demonstrating their effectiveness across binary analysis and comprehension tasks.

The transitions between these steps ensure a seamless progression from foundational alignment to representation refinement and practical application. This structured pipeline integrates the learning stages cohesively, as depicted in Figure 2 and aligned with the corresponding text sections.

## 3 Experimental Design

In this section, we introduce the experimental design for the evaluation of CONTRABIN. Specifically, we design and conduct four experiments in total: one assessment of the pre-training of CONTRABIN, and three others for evaluating the performance of CONTRABIN's embeddings using indicative downstream tasks relating to binary code analysis. We introduce these experiments to answer the following research questions:

- RQ1: How accurately can the embeddings provided by CONTRABIN perform in the tasks of binary code analysis? Does CONTRABIN outperform state-of-the-art pre-trained language representation models evaluated by classification metrics?

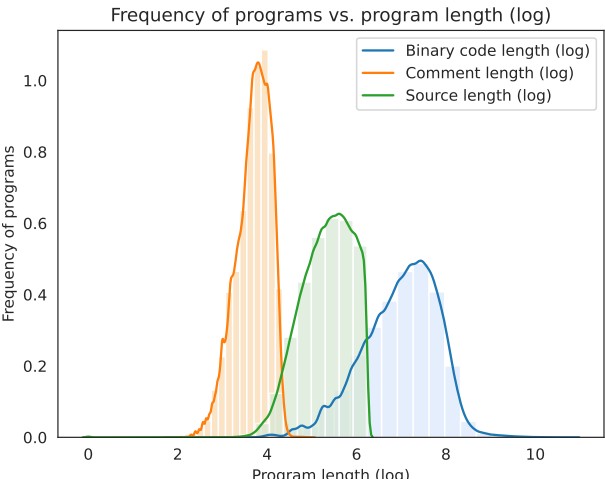

Figure 5: Length distribution of the source code, binary code, and comments in the preprocessed dataset. Comments are skewed to shorter lengths.

- RQ2: How well do the embeddings provided by CONTRABIN in the task of binary code comprehension? Is it better than state-of-the-art pre-trained language representation models evaluated by NLP metrics and human perception?
- RQ3: How effective is the linear and non-linear intermediate contrastive learning in CONTRABIN, and what are the advantages and areas for improvement?

Before we evaluate each RQ in turn, we first introduce the experimental configuration, including the dataset for training and evalution, as well as the experimental procedures and settings.

## 3.1 Training Design

To train CONTRABIN, we must build a dataset consisting of three program representations: source code, compiled binary code, and source code comments or summaries, and train CONTRABIN to generate improved embeddings for downstream tasks. We obtain each program representation as follows:

**Source code:** To obtain source code, we use a widely-used public dataset called AnghaBench (Da Silva et al., 2021) as our source data for training CONTRABIN[4]. Specifically, we adopt the benchmark dataset that contains 1 million single-function C files extracted from programs and mined from popular GitHub repositories. This dataset serves as the source code representation in our method to incorporate semantic variety into the training process.

**Binary code:** After obtaining the source code from AnghaBench, we compile the code snippets in AnghaBench using Clang (specifically, LLVM[5]) to generate the corresponding assembly code. We choose LLVM assembly for platform transparency — many static and dynamic analyses exist for LLVM, and the Clang infrastructure supports many language backends from LLVM bitcode. However, in practice, any straight-line assembly language could fit the requirements of CONTRABIN (i.e., the contrastive learning framework).

**Comments:** During pre-analysis, we found most comments in real-world source code are not globally informative. Specifically, human-written comments can be random in content (e.g., they may contain copyright notifications, random snippets of old code, or unstructured explanations), or contain partial information about the current location of code statements — source code comments in general are not always descriptions of semantics of a program's complete source code. Therefore, we adopt an Encoder-Decoder CodeT5 (Wang et al., 2021) model to automatically generate a single comment for each snippet of source code in our dataset.

---

[4]AnghaBench: http://cuda.dcc.ufmg.br/angha/home
[5]LLVM Compiler Infrastructure: https://LLVM.org/

Figure 5 shows the length distribution of the processed dataset derived from AnghaBench. The overall comment length distribution is quite different from both source and binary code since comments are considered a high-level abstraction of information using natural languages (as opposed to structured programming or assembly languages). We also compute the 90th percentile of length and find the results for source code, binary code, and comments to be 422, 2853, and 63, respectively.

We use two Nvidia A40 GPUs during model pre-training and follow the parameter settings of Simple-CLIP (Shariatnia, 2021). We set the random seed as 42 in the pre-training for reproduction. To better evaluate the performance and robustness, we also train two versions of our model, CONTRABIN-PCL and CONTRABIN. For CONTRABIN-PCL, we set 10 epochs for primary contrastive learning only. For CONTRA-BIN, we use 10, 10, and 10 to improve the overall model efficiency on binary code analysis, and 10, 30, 30 to enhance the general model performance on binary code comprehension.

## 3.2 Evaluation Design

Our evaluation of each research question considers different tasks and perspectives. For each of our three downstream tasks, we choose a publicly available dataset (and corresponding reference in the literature). We preprocess each dataset so that they can leverage the pre-trained embeddings from CONTRABIN. The evaluation datasets include different source code and their compiled assembly. We describe each task and associated dataset in detail below.

**POJ-104** For the binary functional algorithm classification task (RQ1: Downstream Task 1), we adopt POJ-104 from the CodeXGLUE dataset (Lu et al., 2021). In an Open Judge (OJ) system, students submit their solutions to a programming problem, and the OJ system judges whether the code can run successfully on all available test suites. In this way, the OJ system aims to improve the programming skills of users. From this dataset, we can find other programs that perform the same task as an input program (e.g., programs can be classified as bubble sort vs. heap sort. vs. Fibonacci sequence, etc.).

The dataset includes programming problems and verified source code solutions. There are 104 programming problems in POJ-104, with 500 examples each. The dataset is categorized into train/development/test sets with non-overlapping program problem labels. Thus, we can view this dataset as containing 104 different classes in which to classify an input program. While the original POJ-104 dataset used source code structural information, we adapt it for classifying input binaries.

In our experiments, we compile each solution in POJ-104 to LLVM assembly code, keeping the functionality label unchanged, as shown in Table 2. This change transforms POJ-104 into an assembly code dataset suiting our binary analysis interests. We fine-tune each model with 2 epochs and a block size of 400. We use training and validation batch sizes of 32 and 8, respectively, and choose learning rate to be 2e-5 and maximal gradient normalization to be 1. In all fine-tuning processes, we use the default random seed of 123456.

**DIRE** For the binary function name recovery task (RQ2), we chose the DIRE dataset used to train DIRTY model(Lacomis et al., 2019), a recent machine learning model augmenting decompiler outputs with variable type and name predictions. The full dataset contains around 1 million functions spread across 75,656 binaries that were mined from public GitHub repositories using GHCC[6]. With the help of a decompiler, each binary is lifted into equivalent C pseudocode and processed into a list of lexemes and an implementation-defined format for enumerating source-level variable information. Because the DIRE dataset is a preprocessed dataset that does not conform with CONTRABIN's expected inputs, we instead obtained the list of GitHub repositories that were used to construct the DIRE dataset and compiled each project ourselves. We modified GHCC to save the intermediate LLVM bitcode objects produced during compilation and disassemble each bitcode object into readable LLVM assembly using the LLVM disassembler.

In our experiments, we select function names with number of functions larger than 200, and remove some function names that are not specific to a type of function (i.e., main, _, and __list_add) to make sure the model is trained on meaningful data. We further strip all LLVM files of their original function names (i.e., @TESTFUN0 as the name of the first function in an LLVM file) and measure CONTRABIN's ability to classify these function names. The dataset statistics are shown in Table 2. We fine-tune each model with

---

[6]GHCC: https://github.com/huzecong/ghcc

Table 2: Dataset statistics for algorithmic classification.

| Data Type | POJ-104 (Functionality Classification) | | AnghaBench (Code Summarization) | |
|---|---|---|---|---|
| | # Problems | # Examples | # Unique Sum | # Examples |
| Train | 64 | 14,614 | 14,700 | 16,383 |
| Dev | 16 | 5,079 | 902 | 910 |
| Test | 24 | 8,102 | 890 | 911 |
| | DIRE (Function Name Recovery) | | AnghaBench (Reverse Engineering) | |
| Data Type | # Names | # Examples | # Unique Source | # Examples |
| Train | 91 | 49,933 | 15,937 | 16,383 |
| Dev | 91 | 2,774 | 909 | 910 |
| Test | 91 | 2,775 | 903 | 911 |

5 epochs and a block size of 256. We use training and validation batch sizes of 8 and 16, respectively, and choose learning rate to be 2e-5 and maximal gradient normalization to be 1. In all fine-tuning processes, we use the default random seed as 123456.

**AnghaBench** For the binary code summarization and reverse engineering tasks, we use the AnghaBench test set (Da Silva et al., 2021) during pre-training, which includes code never been seen by the model. Specifically, we only select the `main` function in LLVM code and its paired source code for fine-tuning the code summarization and reverse engineering tasks. To better fit the capacity of pre-trained language models, we further truncate the length of binary code by only selecting the first 512 instructions. The dataset statistics are displayed in Table 2. In this paper, we evaluate the model's ability to translate and summarize binary code using the validation set and analyze its embedding using the test set.

In our experiments, we defined the input length for both the summarization and reverse engineering tasks as 512, and the output length for summarization and translation as 32 and 512, respectively. We use training and evaluation batch size of 16 and beam size of 5. We set the number of epochs for the summarization task to 5 and the batch number for the reverse engineering task to 20000. For reproducibility, the random seed is set to 42 for both tasks.

## 4 Experimental Results

In this section, we present the results of our evaluation, addressing each of our research questions. We provide a detailed analysis of the data, discussing how it supports or challenges our hypotheses. Specifically, we examine the effectiveness of our proposed approach in integrating binary code into large-scale pre-training models and incorporating rich information from source code and comments into binary code. We also evaluate the generalizability of our model across different downstream tasks in binary code analysis.

### 4.1 Embedding Analysis

We present a case study to analyze the binary code embeddings generated by our model. While binary codes in the AnghaBench test set are unique, semantically related codes can naturally cluster together in the embedding space due to shared functionality or structural patterns. This clustering is an expected property of a well-trained model and demonstrates its ability to encode meaningful relationships among binary codes. This analysis provides an initial qualitative evaluation of the embedding behavior, offering insights into how our model encodes these relationships. While this analysis highlights trends in embedding quality, it is not intended to serve as definitive evidence, and we provide more robust quantitative evaluations in subsequent sections.

To ensure a fair and direct comparison, all baseline models presented in our evaluation were subjected to the exact same fine-tuning protocol as CONTRABIN. This standardized procedure includes the use of the identical triplet dataset, data splits (training, validation, and test), optimizer (AdamW), learning rate

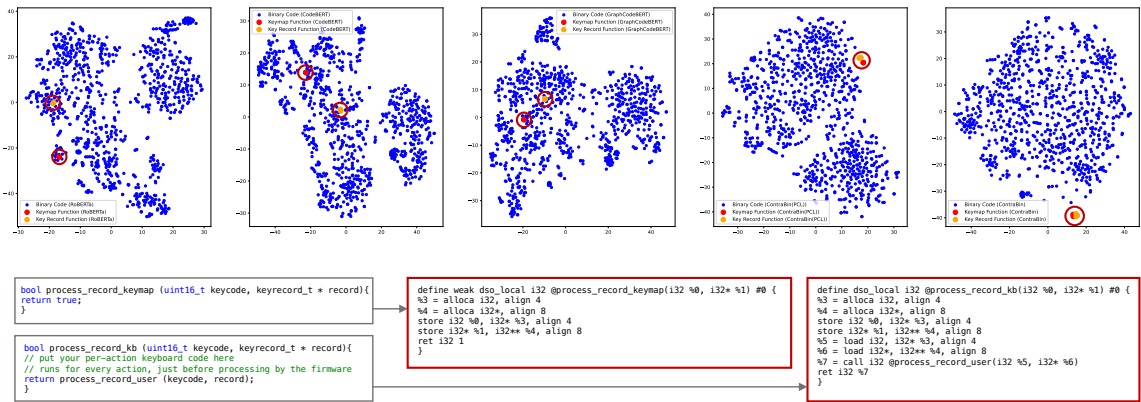

Figure 6: A t-SNE visualization of binary code embeddings from the AnghaBench test set. Each plot shows the overall embedding space (blue dots) from a different model. The highlighted red and orange dots represent two specific, semantically similar functions (`process_record_keymap` and `process_record_user`) to illustrate semantic alignment. Baseline models like RoBERTa place these similar functions far apart and produce a scattered overall distribution. In contrast, both CONTRABIN (PCL) and the final CONTRABIN model cluster the similar functions closely and organize the overall embedding space into more structured regions, demonstrating an improved ability to capture semantic relationships.

schedule, batch sizes, and early-stopping criterion. By maintaining a consistent experimental setup across all models, we can attribute performance differences primarily to the quality of the pre-trained embeddings themselves.

**Case study: Semantic alignment of binary codes** To evaluate the semantic alignment, we selected two binary functions derived from `process_record_keymap` and `process_record_user`, which share similar input structures and functionality. As shown in Figure 6, embeddings generated by RoBERTa, CodeBERT, and GraphCodeBERT project these functions far apart in the embedding space. This separation indicates that these models struggle to capture the shared semantic characteristics of the two functions. In contrast, embeddings generated by CONTRABIN (PCL) position the two functions closer together, reflecting a better alignment of their shared semantics. CONTRABIN further reduces the distance between these functions, demonstrating its ability to refine binary code embeddings through intermediate contrastive learning.

**General trends in embedding space structure** To analyze how different models organize binary codes in the embedding space, we examined patterns across the AnghaBench test set. Figure 6 illustrates that embeddings generated by RoBERTa, CodeBERT, and GraphCodeBERT exhibit scattered distributions, with no discernible grouping for binary codes derived from similar tasks, such as hash table operations or data processing routines. In contrast, embeddings from CONTRABIN (PCL) display a more structured distribution, where binary codes related to similar functionalities begin to cluster into distinct regions. Embeddings generated by CONTRABIN refine this structure further, consolidating these clusters into a cohesive region while maintaining separation for unrelated binary codes. This visualization reveals distinct patterns in how the embedding spaces produced by different models are structured, offering insights into the organization of binary code representations.

Our evaluation protocol for all downstream tasks strictly adheres to the standards set by the official CodeXGlue benchmark. This includes the mandated use of its specified default random seed of 123456 for all experiments. This decision is crucial for ensuring that our results are fully reproducible and maintain direct comparability with the established body of work evaluated on this framework.

Table 3: Quantitative evaluation of CONTRABIN on binary functional algorithm classification of POJ104 using mean of average precision (MAP), recall (MAR) and F1 score (MAF1).

| Approaches | MAP | MAR | MAF1 |
|---|---|---|---|
| RoBERTa (Liu et al., 2019) | 38.29 | 28.57 | 32.72 |
| CodeBERT (Feng et al., 2020) | 39.99 (+4.44%) | 29.94 (+4.80%) | 34.24 (+4.64%) |
| GraphCodeBERT (Guo et al., 2020) | 39.48 (+3.11%) | 29.38 (+2.95%) | 33.69 (+2.96%) |
| CONTRABIN (PCL) | 37.03 (-3.29%) | 26.99 (-5.53%) | 31.22 (-4.59%) |
| CONTRABIN (Ours) | **43.78 (+14.34%)** | **33.87 (+18.55%)** | **38.19 (+16.71%)** |

Table 4: Quantitative evaluation of CONTRABIN and multiple pre-trained methods on binary function name recovery of DIRE using accuracy, mean of average precision (MAP), mean of average recall (MAR), and mean of average F1 score (MAF) as four key criteria.

| Approaches | Accuracy | MAP | MAR | MAF1 |
|---|---|---|---|---|
| RoBERTa (Liu et al., 2019) | 29.41 | 28.41 | 25.00 | 26.59 |
| CodeBERT (Feng et al., 2020) | 24.94 (-15.20%) | 23.25 (-18.15%) | 20.82 (-16.70%) | 21.97 (-17.39%) |
| GraphCodeBERT (Guo et al., 2020) | 25.95 (-11.76%) | 27.59 (-2.87%) | 22.52 (-9.91%) | 24.80 (-6.74%) |
| CONTRABIN (PCL) | 28.83 (-1.96%) | 28.65 (-0.85%) | 25.83 (-3.35%) | 27.17 (-2.16%) |
| CONTRABIN (Ours) | **33.41 (+13.60%)** | **30.79 (+8.39%)** | **28.14 (+12.58%)** | **29.41 (+10.58%)** |

## 4.2 RQ1: Binary Code Analysis

After analyzing embedding of binary code using all pre-trained models, we start qualitative binary code analysis of CONTRABIN by two downstream tasks: algorithmic functionality classification and binary functiona name recovery.

**Result for POJ-104** Our first downstream task is algorithmic functionality classification for binary code. Recall that, in this setting, we use pre-trained embeddings from CONTRABIN to aid classifying an input binary snippet into one of 104 classes of algorithms (e.g., bubble sort vs. heap sort vs. Fibonacci, etc.).

To evaluate the performance of CONTRABIN in this downstream task, we adopt mean of average precision (MAP), mean of average recall (MAR), and mean of average F1 score (MAF) as three key criteria. We consider only C code that can be compiled to binaries — while most programs compiled successfully, we exclude those that could not be compiled, and report MAP, MAR, and MAF across all 104 classes. For this task and the three subsequent tasks, we used the results of RoBERTa as a baseline for comparison.

Table 3 presents the quantitative evaluation of CONTRABIN against several pre-trained methods. CodeBERT and GraphCodeBERT shows comparable performance to RoBERTa on all three metrics, with a performance improvement of less than 5%. However, CONTRABIN demonstrates a substantial improvement in MAP, MAR, and MAF1 by 14.34%, 18.55%, and 16.71%, respectively. We note that the performance of CON-TRABIN (PCL) is lower than that of RoBERTa. This can be partially attributed to the use of contrastive learning over three different modalities, which can result in the binary code embedding becoming overfitted to the exact embedding of either source code or comments, making the model less effective.

**Result for DIRE** Our second task for binary code analysis is function name recovery, which is transformed into a function name classification task where the models are required to classify binary code into one of the 91 function names based on the settings. To evaluate the model performance, we used the same metric set as in POJ-104, with an additional accuracy metric, and applied all models to compiled LLVM code of the DIRE dataset.

The overall performance, as shown in Table 4, revealed that fine-tuning CodeBERT and GraphCodeBERT resulted in a decrease in performance, making them less effective than the baseline RoBERTa model in all metrics by up to approximately 20%. This demonstrated that current models are unable to generalize when the domain of downstream applications for binary code analysis shifts and may even become ineffective. On

Table 5: Quantitative evaluation of CONTRABIN and multiple pre-trained methods on binary code summarization of AnghaBench using BLEU-4, GLEU-4, ROUGE-2, and exact match (xMatch). Performance differences are calculated relative to GraphCodeBERT.

| Approaches | BLEU-4 | GLEU-4 | ROUGE-2 | xMatch |
|---|---|---|---|---|
| RoBERTa (Liu et al., 2019) | 24.30 (-25.05%) | 25.55 (-22.45%) | 30.16 (-23.52%) | 4.95 (-37.41%) |
| CodeBERT (Feng et al., 2020) | 32.07 (-1.11%) | 32.80 (-0.43%) | 38.83 (-1.57%) | 7.91 (0.00%) |
| GraphCodeBERT (Guo et al., 2020) | 32.43 (Base) | 32.94 (Base) | 39.45 (Base) | 7.91 (Base) |
| CONTRABIN (PCL) | 30.89 (-4.75%) | 31.13 (-5.50%) | 37.02 (-6.15%) | 6.70 (-15.28%) |
| CONTRABIN (Ours) | 34.36 (+5.95%) | 34.82 (+5.71%) | 41.20 (+4.43%) | 9.34 (+18.10%) |

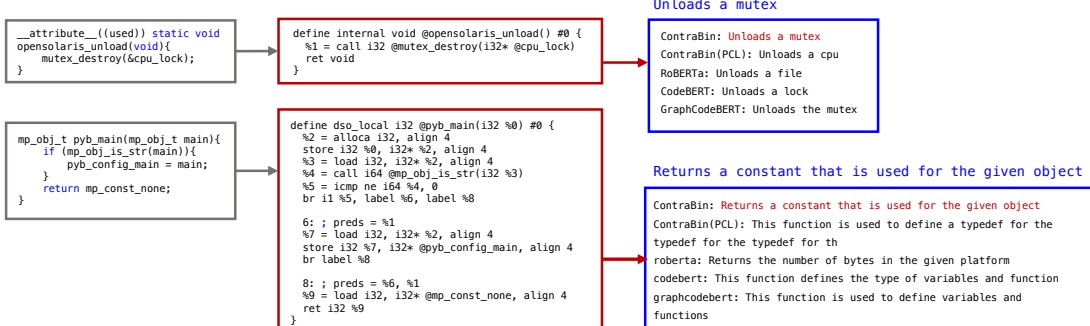

Figure 7: A positive case study on binary code summarization. The first case demonstrates CONTRABIN's ability for semantic completion, while the second showcases its long-term semantic reconstruction capability.

the other hand, CONTRABIN still demonstrated considerable improvement over all three metrics by 13.6%, 8.39%, and 10.58%, respectively, demonstrating the generalizability of our pretrained model on different downstream tasks.

### 4.3 RQ2: Binary Code Comprehension

We further analyze CONTRABIN's ability in binary code comprehension by evaluating its performance on two additional downstream tasks: binary code summarization and binary code reverse engineering. We used the test set of AnghaBench to perform in-domain analysis to assess the effectiveness of our proposed approach.

**Result for AnghaBench (Summarization)** For the binary code summarization task, we evaluate all models using four widely adopted metrics: BLEU-4 (Papineni et al., 2002), GLEU-4 (Mutton et al., 2007), ROUGE-2 (Lin, 2004), and Exact Match (xMatch). As shown in Table 5, CONTRABIN achieves consistent improvements across all metrics compared to the baseline method, GraphCodeBERT. Specifically, CONTRABIN shows improvements of 5.95% in BLEU-4, 5.71% in GLEU-4, 4.43% in ROUGE-2, and 18.08% in xMatch over GraphCodeBERT, showcasing its superior capability in binary code summarization.

Table 6: Quantitative evaluation of CONTRABIN on binary reverse engineering of AnghaBench using BLEU-4, GLEU-4 and exact match (xMatch).

| Approaches | BLEU-4 | GLEU-4 | xMatch |
|---|---|---|---|
| RoBERTa (Liu et al., 2019) | 69.72 | 68.29 | 23.52 |
| CodeBERT (Feng et al., 2020) | 70.37 (+0.94%) | 67.27 (-1.50%) | 23.41 (-0.47%) |
| GraphCodeBERT (Guo et al., 2020) | **71.56 (+2.64%)** | 69.88 (+2.32%) | 24.73 (+5.14%) |
| CONTRABIN (PCL) | 69.73 (+0.02%) | 69.41 (+1.64%) | 23.41 (-0.47%) |
| CONTRABIN (Ours) | 71.14 (+2.04%) | **70.41 (+3.10%)** | **25.16 (+7.01%)** |

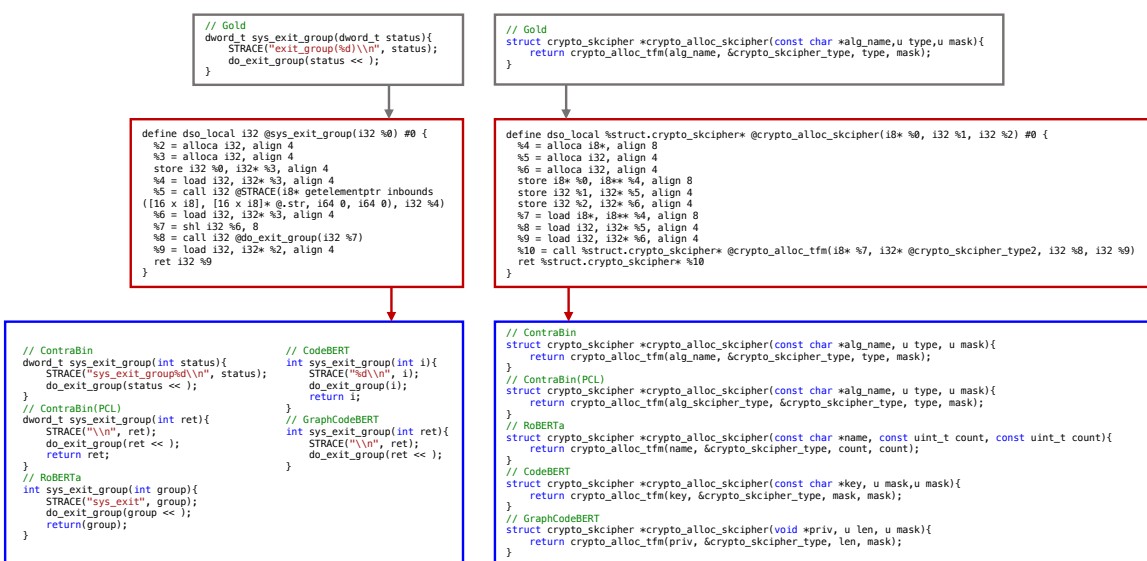

Figure 8: A positive case study on binary reverse engineering. The first case demonstrates ContraBin's ability to control generated content, while the second showcases its ability to maintain semantic consistency.

To further highlight the effectiveness of ContraBin, we present a case study in Figure 7. In the first example, ContraBin successfully captures the functionality of the binary program, while other models diverge to irrelevant objectives, such as "cpu," "file," or "lock." In the second example, other models struggle with summarizing longer binary code. For instance, ContraBin (PCL) generates incomplete sentences, whereas RoBERTa, CodeBERT, and GraphCodeBERT produce either incorrect or overly general summaries. These cases demonstrate ContraBin's ability to maintain long-term semantic consistency and robustness in binary code summarization, enabled by its novel binary embeddings.

**Result for AnghaBench (Reverse Engineering)** For the binary code reverse engineering task, we use similar metrics, except ROUGE, which is designed for summarization only, to evaluate the performance of ContraBin on direct code translation between source code and binary code. As shown in Table 6, ContraBin improves the baseline method by 2.04% in BLEU, 3.10% in GLEU, and 7.01% in xMatch. Compared with GraphCodeBERT, which outperforms ContraBin in BLEU, ContraBin outperforms on the other two metrics (GLEU and xMatch).

We also show a case study in Figure 8. In the first case, ContraBin can maintain the consistency of variable names in the generated source code, while the other methods have issues such as changing variable names and adding incorrect statements. In the second case, the other methods also generate more redundant information. This highlights the ability of ContraBin to generate semantically consistent binary code translations.

## 4.4 RQ3: Analysis and Summary of Model

We can evaluate the effectiveness of different components of ContraBin through the performance of ContraBin (PCL). Our novel model component improves the quality of binary code embedding, as evidenced by our embedding analysis. Our proposed intermediate contrastive learning improves model performance and enhances the robustness and generalizability of the model across various downstream tasks in binary code analysis, as demonstrated by tasks 1 and 2. Furthermore, ContraBin can accurately summarize and translate more binary code while maintaining semantic consistency and long-term dependent coverage, as indicated by tasks 3 and 4. Demonstrated by all four downstream tasks, including two tasks for binary code analysis and two tasks for binary code comprehension, we can confidently conclude that ContraBin enhances the quality of binary code embeddings, leading to improved model performance and greater generalizability.

Table 7: Ablation studies of CONTRABIN on binary functional algorithm classification of POJ104. We report mean average precision (MAP), mean average recall (MAR), and mean average F1 score (MAF1). Absolute scores are shown, with differences from the full CONTRABIN baseline reported in parentheses (pt). Note: Human-written comments provide minimal discriminative signal, resulting in very low absolute performance.

| Ablation Variant | MAP | MAR | MAF1 |
|---|---|---|---|
| Human-written comments | 3.11 (-40.67) | 3.01 (-30.86) | 3.06 (-35.13) |
| Comment removal | 42.76 (-1.02) | 33.23 (-0.64) | 37.40 (-0.79) |
| Anchor removal | 40.96 (-2.82) | 31.07 (-2.80) | 35.34 (-2.85) |
| Interpolation removal | 37.03 (-6.75) | 26.99 (-6.88) | 31.22 (-6.97) |
| CONTRABIN (Full model) | **43.78** | **33.87** | **38.19** |

## 5 Ablation Studies and Model Insights

In this section, we delve into the critical components of our model through extensive ablation studies and comparisons, demonstrating the impact of each design choice on the overall performance. Additionally, we explore the adaptation of our model architecture to contemporary LLMs, particularly the transition from RoBERTa to T5, and how our pretraining strategy enhances the model's understanding of binary code. The findings underscore the importance of each element in our approach and validate the effectiveness of our training methodology.

### 5.1 Ablation Studies

We assess each component of our model through ablation studies. Specifically, we remove three conditions to test the efficiency of each design element: (1) We remove comments as one of the modalities during training, (2) we replace the anchored encoder with a learnable one so that it can be updated during training, and (3) we eliminate both linear and non-linear interpolation from our model design. For the pretraining process, we adhere to the original hyperparameters, training the first and second ablations for 30 epochs and the third ablation for 10 epochs. For the third ablation, we ensure that the model is converged by the end of pretraining. We test the pretrained models on the binary functional algorithm classification of POJ-104, using the same hyperparameters. The results are shown in Table 7.

From the table, it is evident that removing comments, stopping gradients, and eliminating interpolation decrease the MAF1 scores by 2.07 pt, 7.46 pt, and 18.25 pt, respectively. This demonstrates the effectiveness of each component in our model design. The baseline performance observed in these ablation studies is as follows: MAP of 43.78, MAR of 33.87, and MAF1 of 38.19. We will use these baseline performance metrics as a reference in the following sub-analysis to further dissect the contributions of individual components.

### 5.2 Comparison of Human-Written and LLM-Generated Comments in Pretraining

In our study, we explored the impact of using human-written versus LLM-generated comments during the pretraining phase of CONTRABIN. The objective was to evaluate how each type of comment influences the model's ability to comprehend and process binary code. While human-written comments provide a diverse range of insights, they are often inconsistent in length and detail, which may limit their effectiveness in large-scale pretraining scenarios. On the other hand, LLM-generated comments tend to be more consistent and focused, potentially offering a more structured learning experience for the model. The following analysis compares the length distributions and performance outcomes associated with both types of comments.

**Comment length distribution analysis** The analysis of comment length distributions, as depicted in Figure 9, reveals a significant difference between human-written and LLM-generated comments. Human-written comments display a broader spread and variability in length, suggesting diverse approaches to code documentation by developers. On the other hand, LLM-generated comments tend to be shorter and more

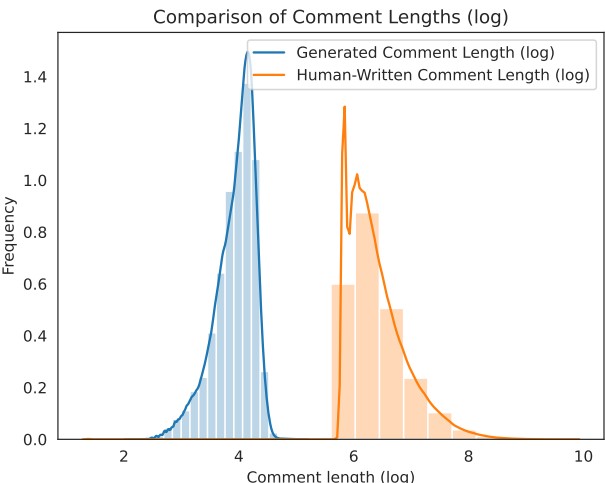

Figure 9: Comparison of log-transformed length distributions for original and extracted comments. The human-written comments exhibit a broader spread indicating diverse lengths, whereas generated comments are predominantly shorter.

consistent in length, indicating a more standardized generation process by the model. These distinctions in comment length and variability may impact the pretraining effectiveness of CONTRABIN. To explore this further, we conducted a series of experiments to compare the performance of CONTRABIN when pre-trained using human-written versus LLM-generated comments.

**Performance comparison** The experiments reveal a striking decrease in performance when CONTRABIN is trained with human-written comments compared to LLM-generated comments. Specifically, Table 7 shows that the use of human-written comments results in MAP dropping from 43.78 to 3.11, MAR from 33.87 to 3.01, and MAF1 from 38.19 to 3.06. These reductions of 40.67 pt, 30.86 pt, and 35.13 pt, respectively, highlight the stark contrast in effectiveness between the two types of comments.

The collapse in performance with human-written comments underscores the critical role of concise and consistent LLM-generated comments in enhancing CONTRABIN's ability to comprehend and analyze binary code. Human-written comments often introduce noise or inconsistencies that hinder representation learning, while the structured nature of LLM-generated comments provides a stronger foundation for binary code analysis. The performance gap can be attributed to the following factors:

- **Redundancy and Verbosity:** Human-written comments often include excessive or redundant details, introducing noise that disrupts semantic alignment between source code, comments, and binary code.

- **Focus on Implementation Details:** Unlike LLM-generated comments, human-written comments tend to emphasize specific implementation nuances rather than summarizing the core functionality, which impairs generalizable learning.

- **Variability in Style and Structure:** Human-written comments exhibit inconsistent styles and structures, creating additional challenges for coherent learning compared to the uniformity of LLM-generated comments.

These findings emphasize the necessity of concise, high-level comments for effective representation learning and highlight the critical impact of training data quality on model performance. Future work should further investigate the interplay between comment quality and model effectiveness to better understand these dynamics.

### 5.3 Multi-Step vs. Multi-Objective Pretraining Approaches

We further explore the impact of different pretraining strategies on the performance of our model. Specifically, we compare the effectiveness of a traditional multi-step pretraining approach with a more integrated multi-objective contrastive learning method. By analyzing these approaches, we aim to understand how the alignment and combination of multiple objectives influence the model's ability to classify binary functional algorithms accurately.

**Multi-Objective Contrastive Learning:** In this study, we introduced a multi-objective contrastive learning approach that incorporates an average of all three losses—binary code, source code, and comments—as a unified multi-objective loss function. This method was designed to encourage the model to simultaneously learn from all available modalities, aiming to enhance its overall ability to classify binary functional algorithms.

**Performance Comparison:** The performance results highlight a dramatic decrease in effectiveness when transitioning from a multi-step pretraining approach to a multi-objective contrastive learning method. Specifically, the multi-objective approach led to a reduction of over 97% across all key metrics—MAP, MAR, and MAF1—compared to the multi-step method. This stark drop underscores the difficulties and risks associated with integrating multiple objectives into a single learning process, particularly when these objectives are not perfectly aligned or when the model struggles to balance the competing losses effectively.

### 5.4 Negative Cases

In this negative case study, we examine a scenario involving function classification in binary code, as illustrated in Figure 10. The original function is a simple program that reads an input, performs a sequence of operations, and prints the result. The code is structured in a way that preserves the flow of data from the input through various function calls to the output. This structure is essential for maintaining the logical integrity of the program, as any deviation in the sequence or function calls could alter the intended behavior.

**CodeT5's Performance** CodeT5 successfully identifies and preserves this structure, correctly mapping the sequence of operations from the source code to the binary function. By doing so, CodeT5 ensures that the original intent of the program is maintained in the binary output, demonstrating its ability to handle the intricacies of binary code classification where functional integrity is paramount.

**ContraBin's Misclassification** In contrast, CONTRABIN misclassifies the function, likely due to its extensive pretraining on natural language and source code semantics. While CONTRABIN's broad pretraining allows it to capture a rich set of semantic relationships, this can lead to overgeneration or misinterpretation when applied to binary code, where precise structural alignment is critical. The model's reliance on semantic cues from source code and comments may introduce biases, causing it to incorrectly infer the function's structure, as seen in this case. This misclassification highlights a potential limitation of CONTRABIN's approach: while it excels in tasks requiring semantic understanding, it may struggle in scenarios where the strict preservation of code structure is necessary for accurate function classification.

### 5.5 Extensibility to Contemporary LLMs

The evolving landscape of LLMs necessitates the adaptation of existing methodologies to newer, more advanced architectures. As LLMs like T5 (Raffel et al., 2020), CodeT5 (Wang et al., 2021) and its derivatives become increasingly prominent, it is crucial to ensure that techniques initially developed for models like RoBERTa can be effectively extended to these contemporary frameworks. This section outlines the key adaptations required to transition from RoBERTa's encoder-only architecture to T5's encoder-decoder structure.

- **Model Architecture Adaptation**: Adapting from RoBERTa to T5 requires handling the transition from an encoder-only architecture to an encoder-decoder framework. This involves modifying the model structure to ensure that the input is properly processed through both the encoder and decoder layers of T5. Specifically, tasks previously handled by RoBERTa's single encoder must now be split

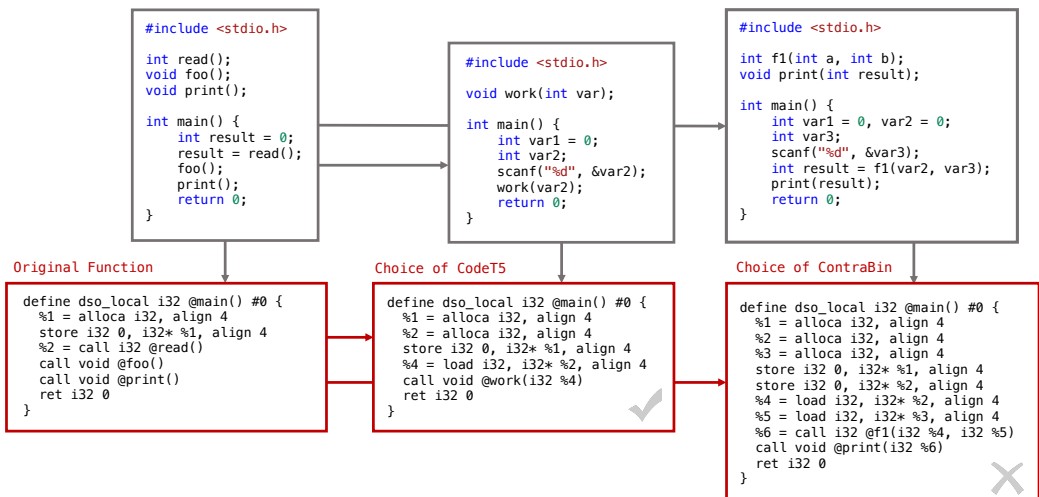

Figure 10: A negative case study on POJ function classification in binary code. The second example shows **CodeT5**'s success in maintaining functional integrity, while the third example highlights a misclassification by **ContraBin**, which overfit to the source and comment information due to contrastive learning.

Table 8: Performance Comparison of CONTRABIN and CodeT5 with T5 on Binary Functional Algorithm Classification for POJ104 (MAP, MAR, MAF1).

| Model | MAP | MAR | MAF1 |
|---|---|---|---|
| T5 (Raffel et al., 2020) | 23.35 | 17.34 | 19.90 |
| CodeT5 (Wang et al., 2021) | 28.82 (+23.42%) | 21.71 (+25.21%) | 24.76 (+24.43%) |
| CONTRABIN (CodeT5) | **30.1 (+28.91%)** | **22.16 (+27.79%)** | **25.53 (+28.34%)** |

between T5's encoder and decoder to fully leverage T5's capacity for generating meaningful output sequences.

- **Tokenizer and Data Processing**: The transition also involves switching from RoBERTa's Byte-Pair Encoding (BPE) tokenizer to T5's SentencePiece tokenizer. This change necessitates revisiting the data preprocessing steps to accommodate the differences in how tokens are generated, including adjustments in token length, padding, and special token handling to ensure compatibility with T5's tokenization requirements.
- **Attention Mechanisms and Loss Function**: The difference in attention mechanisms between RoBERTa and T5 requires revising the attention computation, especially in the forward pass. Additionally, because T5 is designed for sequence-to-sequence tasks, the loss function and output processing need adjustments to align with T5's architecture, ensuring that the model optimizes effectively during training.

For this task, we utilized 8 Nvidia A100 GPUs for CONTRABIN (CodeT5) model pre-training, with the random seed set to 42 to ensure reproducibility. Since the focus was exclusively on binary code analysis rather than comprehension, we adjusted our training strategy accordingly. We reduced the training duration to 40% of the previous rounds to concentrate the model's efforts on this specific task. This reduction was intentionally implemented to optimize the model's performance in binary code analysis, ensuring precision and efficiency without unnecessary overextension.

As shown in Table 8, the CONTRABIN (CodeT5) model with our pretraining method achieves significant improvements over CodeT5. Specifically, our model achieves a MAP of 30.1, which is a 4.45% improvement over CodeT5, a MAR of 22.16 (+2.07%), and a MAF1 of 25.53 (+3.11%). These results underscore the effectiveness of our training methodology in enhancing the model's ability to comprehend and process binary code, demonstrating the adaptability and strength of the T5 architecture.

By addressing the challenges in model architecture, tokenization, and attention mechanisms, we have demonstrated how our approach not only maintains but also enhances its effectiveness in processing binary code across different LLM architectures. This adaptability underscores the flexibility and robustness of our methodology, particularly when applied to state-of-the-art models like T5.

## 6 Related Work

In this section, we highlight key areas of related work, including traditional program analysis techniques and neural models for binary code analysis. Additionally, we discuss the foundational concepts that inspired the design of CONTRABIN, such as large-scale pre-trained embeddings, simplex interpolation, contrastive learning, and graph representation learning. These areas provide a strong basis for understanding how CONTRABIN leverages advancements in program analysis and machine learning to improve binary code representation and comprehension.

**Binary code analysis** Binary code analysis plays an important role in the bigger domain of software analysis and maintenance. For example, tracing execution can help the analysis of the functionality algorithms from binaries (Pierce & Mudge, 1994), reusing profile information can speed up the similarity comparison of frequently executed core code (Wang et al., 2000), incorporating sequences of system calls can help detect software vulnerabilities (Giffin et al., 2004), and combining machine and binary-interface descriptions can assist software reverse engineering (Cifuentes et al., 1999). Traditionally, in programming language and software engineering, researchers have presented various work based on static and dynamic program analysis techniques. For instance, BYTEWEIGHT (Bao et al., 2014) automatically classifies algorithms by functionality from binaries, BINSEC (Djoudi & Bardin, 2015; David et al., 2016) formalizes low-level regions of code to extract semantics on the programs, and BinSide (Aslanyan et al., 2020) uses intermediate representations to conduct cross-platform static analysis.

**Neural code models** In recent years, neural code models have attracted the attention of researchers in software engineering and security. With the assistance of AI (Vaswani et al., 2017), it can improve many source code analysis tasks, such as neural clone detection, neural code similarity comparison, neural code search, etc (Ben-Nun et al., 2018; Shi et al., 2019; Zeng et al., 2021; Ye et al., 2020; Zhang et al., 2022).

For example, SySeVR combines syntax, semantics, and neural vector representation to detect software vulnerabilities (Li et al., 2021), InnerEye adopts techniques in Natural Language Processing (NLP) to compare between samples of binary code (Zuo et al., 2019), GNN-BPE (Guo et al., 2022b) applies Graph Neural Networks (GNN) on binary code by fusing the semantics of Control Flow Graphs (CFG), Data Flow Graphs (DFG), and call graphs, and Bin2Vec (Arakelyan et al., 2021) learns binary code representation via Graph Convolutional Networks (GCN). Compiler chain detection has also emerged as a promising application of machine learning in binary analysis, enabling the identification of the toolchains used to generate binary executables and aiding tasks like provenance analysis and malware detection (Chen et al., 2022a; De Blaere et al., 2023). Additionally, the survey by Marcelli et al. (Marcelli et al., 2022) provides a detailed review of how machine learning techniques are solving the binary function similarity problem, which complements the above-discussed approaches. However, these methods involve complex aggregation of additional binary code representations and have limited generalizability across tasks.

**Large-scale pre-trained embeddings** With the recent success of large-scale pre-trained embedding in AI, research has been conducted on applying similar approaches to code analysis (Ben-Nun et al., 2018; Shi et al., 2022; Jiang et al., 2022; Bertolotti & Cazzola, 2022). In particular, RoBERTa (Liu et al., 2019) serves as the baseline, CodeBERT (Feng et al., 2020) incorporate code generation and identification into training process, GraphCodeBERT (Guo et al., 2020) combines Abstract Syntax Tree (AST) to further improve embedding quality. There are also many recent models designed for source code understanding, including CodeTrans (Elnaggar et al., 2021) and CoTexT (Phan et al., 2021) that incorporate different language modalities and multi sub-tasks for better code representation learning, and CodeT5 (Wang et al., 2021) which proposes an identifier-aware pre-training task to improve embedding distinction. CONTRABIN aims similar goals with these large-scale language representation models but focuses on a more effective and efficient representation for binary code in terms of better semantics

**Contrastive learning** As a recent emerging research direction, contrastive learning can enhance the performance of many computer vision tasks by contrasting samples against each other to learn common properties (Tian et al., 2020; Xiao et al., 2020). For example, MoCo (He et al., 2020) proposes a dynamic dictionary to facilitate contrastive learning, SimSiam(Chen & He, 2021) introduces a Siamese network and stop gradient scheme to prevent collapsing in optimization, and CLIP (Radford et al., 2021) integrates visual concepts and raw text about images to provide much broader source of supervision in multi-modal contrastive learning. As for program understanding, ContraCode (Jain & Jain, 2021) is the first work using contrastive learning in source code analysis by code augmentation and comparison. We design CONTRABIN based on the contrastive learning framework which simultaneously and gradually learns from binary code, source code, and comments within the same program.

## 7    Conclusion

To summarize, we propose a novel approach that fully integrates binary code into the large-scale pre-training model framework and incorporates rich information from source code and comments into binary code. Our experiments demonstrate that our proposed components outperform other approaches in four downstream tasks in binary code analysis and comprehension. We believe that our research work can inspire the AI, software engineering, and security communities to develop new methods that can further improve binary code analysis and comprehension and facilitate binary code applications from various perspectives.

## Data Availability Statement

The data and code used in this study are publicly available on Zenodo at `https://zenodo.org/records/15219264`. The repository includes:

- Preprocessed datasets used for training and evaluation.

- Complete implementation of the CONTRABIN framework, including all scripts for data preprocessing, model training, and evaluation.

- Detailed documentation and configuration files to facilitate reproducibility and ease of use.

Researchers are encouraged to access the repository to replicate our findings or use the provided resources as a foundation for further studies. Any questions or issues regarding the data or code can be directed to the corresponding authors.

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
