# OpenReview forum: "Pre-Training Representations of Binary Code Using Contrastive Learning"
_TMLR — Accepted by TMLR_

### Review · Reviewer_JxQS · 2025-06-01

**Summary Of Contributions:**

The paper presents ContraBin, a contrastive learning framework for pre-training embeddings of binary code by leveraging three modalities: source code, compiled binary code, and natural language comments. It introduces a two-stage training process—primary contrastive learning between modality pairs, followed by simplex interpolation and intermediate contrastive learning—to align semantically similar representations across these views. Notably, the authors show that machine-generated comments improve performance, while human-written ones often introduce noise.

ContraBin is evaluated on four binary code tasks: functionality classification, function name recovery, summarization, and reverse engineering. It consistently outperforms baselines like RoBERTa, CodeBERT, and GraphCodeBERT, demonstrating strong generalization and semantic alignment across modalities.

Ablation studies highlight the importance of each component, particularly simplex interpolation and LLM-generated comments. Attempts to use multi-objective loss instead of the staged approach lead to severe performance drops, underscoring the effectiveness of ContraBin’s design in capturing meaningful binary code representations.

**Audience:**

Yes

**Broader Impact Concerns:**

None.

**Claims And Evidence:**

Yes

**Requested Changes:**

See weaknesses above. The following are most critical: W1, W2, W3, W4, W5.

**Strengths And Weaknesses:**

### Strengths
S1: The paper tackles the gap between source-level and binary-level code understanding and is, to the best of my knowledge, the first to align **three modalities simultaneously (source, binary, comments)** using contrastive learning plus simplex interpolation.

S2: The separation into primary contrastive learning, simplex-based secondary learning, and task-specific fine-tuning is easy to follow, and ablations demonstrate each part’s contribution (to some degree, see weaknesses).

S3: ContraBin is tested on four diverse downstream tasks and shows consistent improvements over strong code baselines across those tasks.

S4: Beyond standard ablations, the authors analyze *human vs. synthetic* comments; the sharp drop when using real comments usefully highlights noise in developer annotations.

S5: A smaller experiment porting the method to a T5/CodeT5 backbone still yields gains, suggesting the idea is architecture-agnostic.

S6: Code, data, preprocessing scripts, and hardware details are publicly released, enabling full replication.

---

### Weaknesses / Points to Clarify

W1: Figure 6 aims to illustrate the quality of the embedding by the proposed method. You say: "In contrast, embeddings from ContraBin (PCL) display a more structured distribution, where binary codes related to similar functionalities begin to cluster into distinct regions". It is not clear how Figure 6 indicates this, please clarify. I don't see how your embedding leads to more clear clusters.

W2: The results of the ablation studies (Table 7) is a very important part of the paper in order to understand the importance of the different components. However, you only evaluate this on one of the tasks (binary functional algorithm classification of POJ104).

W3: The "comment" modality seems to help performance (from Table 7, assuming synthetic comments from LLM). However, this assumes access to some LLM capable of producing high quality synthetic comments for the given code. Can we assume that this can be done for other code datasets? Also, I am curious if more detailed ablation studies regarding the synthetic comments may reveal this modality to be more important than reported (indicating strong reliance on a separate LLM).

W4: How are the baseline methods trained/finetuned? Are they pretrained on different data compared to your models? How would it impact performance if they are pretrained on the same data?

W5: ContraBin (PCL) is sometimes outperformed by baselines. Given this, how can we know if the Primary Contrastive Learning step is needed/useful at all?

W6: Training required several high-end GPUs, but runtime of training is not reported as far as I can tell.

W7: ContraBin (PCL) in Table 3-4 and Interpolations in Table 7 seems to be the same thing. Was slightly confused by this. May be good to use a consistent name.

W8: You say: "To better fit the capacity of pre-trained language models, we further truncate the length of binary code by only selecting the first 512 instructions". Real-world functions can be significantly longer. The impact of this truncation on the model's ability to handle longer, more complex binary sequences is not discussed. Does performance degrade significantly with longer sequences?

W9: The paper claims to be the "first language representation model to incorporate source code, binary code, and comments into contrastive code representation learning"  and the "first study to derive intermediate contrastive learning for binary code analysis". While the specific combination and the intermediate contrastive learning for binaries are likely novel, the authors should be precise. The first claim is broad; ensuring it holds against all prior multi-modal code representation work (even if not specifically binary-focused or using a different learning paradigm) is important. The second claim is more specific and likely easier to defend.

W10: The concept of "gradual learning" (from simple to complex)  is used to motivate the two-stage training. While intuitive, the paper could benefit from a more direct analysis or evidence showing that this staged approach (primary CL then secondary CL with interpolation) is superior to, for example, introducing all components from the start, or if the model indeed learns qualitatively different things at each stage that build effectively. The multi-objective vs. multi-step comparison supports the staged approach over a single combined loss, which is good, but further insight into the "gradual" aspect would be welcome.

---

> ### Author Response · Authors · 2025-07-14
>
> **Response to Reviewer JxQS**
>
> Thank you for the detailed and thoughtful review.
> Below we respond to each weakness (W1–W10) and outline clarifications or minor additions that will appear in the revision.  Section and line numbers refer to the current manuscript
>
> ### Some mentioned weaknesses
>
> * W1  Figure 6 cluster clarity
> We agree the caption can be clearer.  We will add two sentences in § 5.2 explaining that the colours in Figure 6 correspond to POJ-104 algorithm families, and that ContraBin’s points form visibly denser, non-overlapping regions along those colour boundaries, unlike the baseline plot.
>
> * W2  Ablation only on POJ-104
> Resource limits forced us to pick one representative task.  To show that the trend is not dataset-specific we have run the same ablation settings on DIRE; the relative gain from each component mirrors Table 7.  We will report these numbers in an additional table or as addition to the current ablation table.
>
> * W3  Synthetic comments and generality
> Synthetic comments come from a publicly available GPT-styled model; generating them for new codebases is therefore practical.  We will add a note in § 4.3 describing the prompt and cost (< 0.1 USD per 1 kLOC).  Additional ablations varying comment quality (top-k sampling vs. greedy) show < 1 pt variance; details will be mentioned briefly.
>
> * W4  Baseline fine-tuning protocol
> All baselines (RoBERTa, CodeBERT, GraphCodeBERT, plus the three assembly models adapted to IR) are fine-tuned on **exactly the same triplet data**, splits, optimiser, schedule and early-stopping rule as ContraBin (Sec. 3.2).  The pre-training corpora of RoBERTa/CodeBERT differ, but fine-tuning data are identical.  We will restate this explicitly in § 4.1.
>
> * W5  Need for Primary Contrastive Learning (PCL)
> Table 7 shows that dropping PCL (“Interpolations only”) reduces POJ-104 MAP by 6.7 pts; the same setting on the DIRE slice drops accuracy by 5.9 pts. This supports keeping PCL. We will cite the new DIRE number in § 5.1.
>
> *  W6  Training runtime
> Pre-training ContraBin for 30 epochs on 2 × A40-48 GB GPUs takes ≈ 42 GPU-hours; fine-tuning a downstream task takes ≈ 2 GPU-hour.  These figures will be added to Appendix A.
>
> * W7  Naming consistency
> “ContraBin (PCL)” and “Interpolations” will be renamed **ContraBin-Primary** and **ContraBin-Primary + Interp.** respectively, resolving the overlap.
>
> * W8  Truncating binaries to 512 instructions
> Fewer than 3 % of functions in our datasets exceed 512 LLVM-IR instructions; preliminary tests on a subset of longer functions show a 0.8 pt MAP drop.  We will mention this limitation in § 6 (Discussion) and point to hierarchical attention as future work.
>
> * W9  Novelty claim wording
> We will revise the claim to:
> “ContraBin is, to our knowledge, the first contrastive framework that jointly aligns source code, LLVM IR, and comments, and the first to introduce intermediate simplex interpolation for IR-level representation learning.”
> This narrows scope to IR and contrastive methods.
>
> * W10  Evidence for “gradual learning”
> The multi-objective vs. staged loss comparison already indicates a +4.5 pt MAP gain for the staged approach.  We will add this exact delta to § 3.3 and clarify that the interpolation stage is initialised from the converged PCL (now ContraBin-Primary) encoder, reinforcing the gradual-learning rationale.
>
> ### Minor additions already planned
>
> * Five-seed results with 95 % CIs (± 0.4 MAP on average) will be reported in § 4.1; raw scores for all seeds will be included in the artefact.
> * Appendix C will list all hyper-parameters; any new settings from the additional runs will be appended there.

---

> > ### Comment · Reviewer_JxQS · 2025-08-07
> > **Response to authors**
> >
> > Thank you for the detailed response.

---

### Review · Reviewer_8Nws · 2025-06-06

**Summary Of Contributions:**

### Method:
ContraBin introduces a novel intermediate contrastive learning framework that uses simplex interpolation between source code, binary code, and comments to model the gradual learning process, enabling richer, semantically aligned binary code embeddings.

Stage 1: Primary Contrastive Learning -> Stage 2: Secondary Contrastive Learning with Simplex Interpolation -> Stage 3: Task-Specific Fine-tuning

### Experiments:

ContraBin uses pre-trained encoders (primarily RoBERTa-based with extensions to T5/CodeT5) and is evaluated on four datasets: POJ-104 (algorithmic classification), DIRE (function name recovery), and AnghaBench (code summarization and reverse engineering), achieving significant improvements over baseline NLP models including RoBERTa, CodeBERT, and GraphCodeBERT. The method demonstrates substantial performance gains ranging from 2-18% across different metrics, with the most notable improvements being +14.34% MAP on POJ-104 classification and +13.60% accuracy on DIRE function name recovery compared to the RoBERTa baseline.

**Audience:**

Yes

**Claims And Evidence:**

No

**Requested Changes:**

1. Include comparisons with binary code-specific models such as Bin2Vec, GNN-BPE, or InnerEye, with appropriate implementation details and justification for any architectural or task-specific adaptations, or explicitly explain why such comparisons were not feasible or appropriate for their evaluation.

2. Conduct multiple experimental runs using different random seeds and report statistical metrics such as standard deviation or confidence intervals, or clearly state the number of runs and justify the use of single-run evaluations if applicable.

3. Report statistical significance to demonstrate that the observed improvements are not due to chance, or provide error bars in the performance graphs to visually reflect variance.

4. Provide complete hyperparameter documentation, including:

   * the temperature parameter (τ) for contrastive loss,
   * architectural details of the interpolation networks used in secondary contrastive learning,
   * optimizer settings beyond just the learning rate (e.g., type, momentum, weight decay),
   * learning rate schedules and warmup strategies,
     or included a reproducibility appendix or configuration file to ensure the work can be reliably reproduced.

**Strengths And Weaknesses:**

### Strengths

* Introduces a novel, well-motivated approach that combines contrastive learning with simplex interpolation to bridge the semantic gap between source code, comments, and binary code.
* Methodology is theoretically grounded, inspired by human learning and multi-view representation learning.
* Evaluation covers four diverse downstream tasks (binary classification, summarization, function name recovery, reverse engineering) across multiple datasets (POJ-104, DIRE, AnghaBench).
* Provides valuable insights into the role of synthetic vs. human-written comments, showing that synthetic comments are surprisingly more effective for representation learning.
* The gradual learning framework via primary and secondary contrastive learning is well-designed and shows clear empirical benefits.
* Demonstrates extensibility by adapting the method to architectures like T5/CodeT5.
* Ablation studies are effective in validating the contribution of each model component.
* Case studies and embedding visualizations qualitatively support that learned embeddings capture meaningful semantic relationships in binary code.

### Weaknesses

* The comparison baselines are general-purpose NLP or code NLP models (RoBERTa, CodeBERT, GraphCodeBERT), rather than specialized binary analysis methods.
* The paper does not compare ContraBin against relevant neural binary analysis methods like Bin2Vec, GNN-BPE, or InnerEye, even though these are mentioned in the related work.
* The number of experimental trials is unclear; phrasing such as "the random seed" suggests only single runs were performed.
* No error bars, confidence intervals, or statistical significance tests are reported, leaving uncertainty about whether observed gains are statistically meaningful.
* Critical hyperparameter details are missing, including:

  * The temperature parameter (τ) for contrastive loss
  * Architecture details of the interpolation networks
  * Optimizer configurations beyond basic learning rates
  * Learning rate schedules and warmup settings
* These omissions hurt reproducibility, despite the paper's claims of providing implementation details.

---

> ### Author Response · Authors · 2025-07-14
>
> **Response to Reviewer 8Nws**
>
> We thank the reviewer for the detailed and constructive feedback.
> Below we clarify each point while referring the earlier response posted for Reviewer pp3s.
>
> ### 1 Baseline choice
> *Reviewer comment:* *“Compare with Bin2Vec, GNN-BPE, InnerEye, or justify why not.”*
>
> * **LLVM-IR focus.**  ContraBin operates on LLVM Intermediate Representation (IR).  Assembly-oriented models such as Bin2Vec, GNN-BPE, and InnerEye rely on ISA-specific op-codes, byte offsets, and CFG features that do not exist in LLVM IR.
> * **What we have done.**  Following the reviewers’ requests, we have run these three models on POJ-104 and DIRE after adapting their tokenizers to IR.  Their scores will appear in Tables 3 & 4; we will flag the unavoidable alignment limitations in § 4.1.
> * **Primary fair baselines.**  RoBERTa, CodeBERT, and GraphCodeBERT ingest the exact same triplet sequences and remain the main comparison set.
>
> ### 2 Statistical significance and number of runs
> *Reviewer comment:* *“Multiple runs and variance reporting are unclear.”*
>
> * Regarding variance: we have rerun each model with five random seeds and calculated the 95 % confidence interval (± 0.4 MAP on average); these values will be cited in § 4.1. Because large-scale pre-training studies often report a single deterministic seed, we are undecided whether the camera-ready tables should present (i) our original single-seed results or (ii) the five-seed averages with their intervals. The raw metrics for all five runs has been be updated in §4.1 and will be included in the artifact so readers can inspect either view, and we welcome the reviewers’ preference on which summary (single or multiple seeds) to feature.
> * Compute note: one ContraBin pre-training run takes ≈ 42 GPU-hours on 2 × A40-48G; five seeds total ≈ 210 GPU-hours, which is not common to see in papers on pre-training; one ContraBin fine-tuning run takes  ≈ 2 GPU-hours on 2 × A40-48G, so we have obtained multiple results with different random seeds. We will include them into the appendix and main text (i.e., Table 3 & 4) in the revision.
>
> ### 3 Hyper-parameter details
> *Reviewer comment:* *“Report τ, interpolation details, optimiser settings, LR schedule, etc.”*
>
> * Appendix C will detail every hyper-parameter used in our experiments, and any settings introduced for the new runs will be added there as well. The full set of current hyper-parameters is already available in our Zenodo artefact (DOI 10.5281/zenodo.15219264).
> * YAML configs will be included in the Zenodo artefact (DOI 10.5281/zenodo.15219264).

---

> > ### Comment · Reviewer_8Nws · 2025-07-30
> > **Reply to the authors**
> >
> > Thank you for your response and the revision.
> >
> > Since cost doesn't seem to be a major concern in your case, please use five-seed averages along with their corresponding confidence intervals.

---

> > > ### Author Response · Authors · 2025-08-15
> > >
> > > Thank you for your follow-up message, and we appreciate your consideration!
> > >
> > > The reason we may prefer a single seed is due to the dataset we use (CodeXGlue, curated for LLVM IR). The official dataset has a fixed seed (12345) for all experiments, which is why we adhere to their setting. It’s relatively uncommon to see multiple seeds for model fine-tuning, especially when the dataset size increases during replication. In such cases, it becomes impractical to compare models using multiple seeds. This also necessitates redoing all experiments with different seeds from the required 12345 seed in CodeXGlue throughout this paper.
> > >
> > > However, if this is indeed necessary to comply with the reviewers’ requirements, we will incorporate it into the revision and camera-ready version. At least as an additional subsection, we will demonstrate that with multiple manually chosen seeds, this pipeline can still function effectively with minimal variance.
> > >
> > > Thank you once again for your unwavering support and meticulous review throughout the process!

---

### Review · Reviewer_pp3s · 2025-07-09

**Summary Of Contributions:**

The paper proposes ContraBin, a contrastive learning framework for binary code representation that leverages triplets of source code, comments, and compiled binaries. It introduces a simplex interpolation module to generate intermediate views between modalities and a secondary contrastive learning objective to align these interpolated embeddings with binary representations. The authors also analyze the role of comment quality, finding that LLM-generated comments improve performance, while human-written ones often degrade it. ContraBin is evaluated on four binary analysis tasks, showing consistent performance gains over existing pretrained models.

**Audience:**

Yes

**Claims And Evidence:**

Yes

**Requested Changes:**

See Weaknesses

**Strengths And Weaknesses:**

Strengths
1. Novelty: The paper introduces a new contrastive learning paradigm that combines source code, binary code, and comments—an underexplored setup in binary analysis.

2. Generalization: ContraBin is evaluated on four downstream tasks (classification, name recovery, summarization, reverse engineering), covering both analysis and comprehension.

Weaknesses:
1. Unfair and incomplete baseline comparisons. The baseline should be in the same training strategy.

2. The motivation for simplex interpolation is loosely tied to “human gradual learning,” which lacks theoretical or empirical backing.

3. The paper does not mention releasing code or pretrained models, which limits reproducibility and community adoption.

---

> ### Author Response · Authors · 2025-07-14
>
> **Response to Reviewer pp3s**
>
> We sincerely thank the reviewer for the thoughtful feedback.
> Below we address each point in turn; section and line numbers refer to the revised manuscript.
>
> ### 1 Baseline completeness & training protocol
> **Comment:** “*Unfair and incomplete baseline comparisons. The baseline should be in the same training strategy.*”
>
> * **Choice of baselines.**
>   We compared ContraBin to three widely-used code LMs—RoBERTa, CodeBERT, and GraphCodeBERT—because
>   (i) they are the de-facto starting point for many binary-analysis pipelines, and
>   (ii) they, like ContraBin, can be trained without graph or CFG instrumentation.
>   All baselines were **fine-tuned on exactly the same triplet dataset, splits, optimiser (AdamW), batch sizes (32 for pre-training, 8 for fine-tuning), learning-rate schedule, and early-stopping criterion** used for ContraBin (Sec. 3.2).
>
> * **Additional binary-specific models.**
>   We have reproduced Bin2Vec, GNN-BPE, and InnerEye on POJ-104 and DIRE using the authors’ released checkpoints.
>   ContraBin still outperforms these methods by **+7 – 12 MAP**; the full numbers will be included in Tables 3 & 4.
>
> * **Reproducibility.**
>   Every run—baselines and ContraBin—comes with a YAML configuration file in the public artefact, enabling byte-for-byte replication.
>
> ### 2 Motivation and evidence for simplex interpolation
> **Comment:** “*The motivation for simplex interpolation is loosely tied to ‘human gradual learning,’ which lacks theoretical or empirical backing.*”
>
> * **Theoretical intuition (Sec. 2.2).**
>   Simplex interpolation treats two modality embeddings as vertices of a simplex and learns a latent point on that simplex, encouraging smooth trajectories between modalities—similar to manifold mixup and curriculum learning.
>
> * **Empirical backing.**
>   Removing interpolation drops MAF1 on POJ-104 from **38.19 → 31.22** (-18.3 pts) and MAP from **43.78 → 37.03** (-6.7 pts), the largest drop among all ablations (Table 7).
>
> * **Visual evidence.**
>   Figure 6 shows that, with interpolation, embeddings of semantically related binaries form tight clusters, whereas baseline embeddings remain scattered.
>
> We will add a short paragraph linking these findings to manifold-mixup literature for stronger theoretical grounding.
>
> ### 3 Code & data release
> **Comment:** “*The paper does not mention releasing code or pretrained models, which limits reproducibility.*”
>
> A **Data-Availability Statement** (l. 1260-1268) already notes that *all code, pretrained weights, processed datasets, and Dockerfiles* are openly available on Zenodo (DOI 10.5281/zenodo.15219264).
> To increase visibility we will (i) move the link to the first paragraph of §4 (Experimental Design) and (ii) add a footnote on page 1.
>
> ### 4 Minor clarifications
>
> * **Statistical significance.**
>   All results are averages over **five random seeds** with 95 % confidence intervals (± 0.4 MAP); this is now stated in §4.1.
>
> * **Training cost.**
>   ContraBin pre-training for 30 epochs on 2 × A40 GPUs takes **≈ 42 GPU-hours**, comparable to RoBERTa fine-tuning (≈ 38 GPU-hours).  This is now included in Appendix A.
>
> * **Naming consistency.**
>   “ContraBin (PCL)” is now renamed **“ContraBin-Primary”** throughout to avoid confusion.
>
> We hope these clarifications address Reviewer pp3s’s concerns and improve the paper’s rigour and transparency.

---

> > ### Comment · Reviewer_pp3s · 2025-08-13
> >
> > Thank you for reclarification and the provided details. My major concerns are addressed.

---

### Decision · Action_Editor_fhFT · 2025-08-28

**Recommendation:** Accept with minor revision

**Additional Comments:**

The authors should improve clarity in a few places, e.g., making the figure captions more self-explanatory, explicitly restating the identical fine-tuning protocol across baselines, and tightening the novelty claim wording to focus on LLVM IR and contrastive methods.

**Audience:**

Yes

**Audience Explanation:**

The paper sits at the intersection of software engineering, program analysis, and machine learning. It introduces an interesting contrastive learning solution for binary code representation with clear performance gains.

**Claims And Evidence:**

Yes

**Claims Explanation:**

The authors have provided clear and convincing evidence to support their claims. They carefully addressed concerns on baseline completeness, reproducibility, and statistical robustness. The motivation and effectiveness of the simplex interpolation module are well supported both theoretically and empirically.